# Effect of Silver Diamine Fluoride and Potassium Iodide Solution on Enamel Remineralization and Discoloration in Artificial Caries

**DOI:** 10.3390/ma15134523

**Published:** 2022-06-27

**Authors:** Ko Eun Lee, Munkhulzii Erdenebulgan, Chung-Min Kang, Hoi-In Jung, Je Seon Song

**Affiliations:** 1Department of Pediatric Dentistry, Kyung Hee University Dental Hospital, Seoul 02447, Korea; olivedlr@naver.com; 2Department of Pediatric Dentistry, College of Dentistry, Yonsei University, Seoul 03722, Korea; steve.bulgaa@gmail.com (M.E.); kangcm@yuhs.ac (C.-M.K.); 3Department of Preventive Dentistry & Public Oral Health, College of Dentistry, Yonsei University, Seoul 03722, Korea

**Keywords:** silver diamine fluoride, potassium iodide, NaF varnish, enamel remineralization, discoloration

## Abstract

Silver diamine fluoride (SDF) is a strong fluoride agent for caries control, remineralization, and reducing the incidence of dental caries. This study used 38% SDF with potassium iodide (KI) on enamel remineralization and compared it with the currently used 38% SDF and 5% sodium fluoride (NaF) varnish to treat artificial caries. Bovine incisors were demineralized for 120 h and randomly distributed for treatment by the remineralization agents for 120 h as follows: control (*n* = 15), SDF/KI (*n* = 15), SDF (*n* = 15), and NaF (*n* = 15). Colorimetric analysis was performed using the ΔE value. The Vickers microhardness (VHN) was determined to evaluate the enamel remineralization of the specimens. Polarized light microscopy (PLM) and scanning electron microscopy (SEM) were performed to confirm the surfaces precisely and histologically. SDF/KI caused no significant difference in discoloration between the control and NaF groups. There was also no significant difference in the VHN after remineralization, but SDF/KI exhibited the highest mean microhardness. PLM revealed that SDF/KI had the greatest remineralization ability. In conclusion, SDF/KI is effective for dental enamel remineralization even with KI, which reduces discoloration.

## 1. Introduction

Dental caries is one of the most prevalent infectious diseases that occurs when the demineralization–remineralization functions are out of balance [1]. The progressive loss of mineralization leads to cavitated dental caries, which requires invasive intervention treatment such as drilling, filling, and replacing with foreign materials [2]. However, non-invasive and non-restorative treatments, such as sealing sticky pit and fissure, and remineralization with CPP-ACP and fluorides, are very effective and widely used [3]. Especially, the fluoride is the easiest and most inexpensive method, and there are various products (e.g., fluoride gels, fluoride mouthwashes, fluoride toothpastes, fluoride foams, and fluoride varnishes) to prevent dental caries and delay the progression of early dental caries [4,5]. Among them, pediatric dentists recommend the use of fluoride toothpaste twice daily coupled with the periodic professional application of fluoride varnish or gel [6].

Silver diamine fluoride (SDF) was first introduced 80 years ago as a formulation containing ionic silver, ammonia, water, and fluoride; its use was approved by the US FDA in 2014 [7]. It is effective for reducing the progression of dental caries in primary, permanent teeth and in deep coronal cavities, as well as for mitigating root caries in elderly patients [8,9] and alleviating the symptoms of sensitivity and sore teeth through the obturation of dentinal tubules [10]. Various studies are underway to investigate the role of SDF in inhibiting the spread of oral bacteria in dentinal tubules. A study reported that SDF reduces the amount of *S. mutans* [11].

However, a critical disadvantage of a single application of SDF is the discoloration of teeth. Yee et. al. reported that enamel and dentin, including carious tissues, are decolorized into dark brown or black after SDF application [12]. The greater the extent of demineralization, the more the silver ions are absorbed, thereby increasing discoloration and the distinction between affected and sound tissues [13]. The stain remains over time and can only be removed through physical methods [14].

Recently, the supplementation of SDF with potassium iodide (KI) was proposed to address the issue of discoloration [15]. It is hypothesized that KI will be extremely beneficial if it can prevent the staining associated with SDF without reducing its effectiveness in preventing caries [16]. Riva Star (SDI Limited, Bayswater, Australia) is one example of such a product, where the excess silver ions in the SDF solution combine with the KI ions to generate silver iodide, which reduces SDF-induced discoloration [17]. There have been studies evaluating the effect of KI addition on dentin and oral bacteria, as well as studies on the bonding strength between the dentin and restoration of teeth treated with SDF/ KI. However, studies on the remineralization effect limited to enamel are still lacking.

Thus, little is known about the effect of SDF/KI on the enamel surface itself, and there is a lack of comparative studies between SDF/KI and other fluoride materials. Therefore, the primary purpose of this study was to evaluate the effect of 38% SDF and KI on enamel remineralization and compare it with that of the currently used 38% SDF and 5% sodium fluoride (NaF) varnish, to treat artificial caries. The secondary purpose of this study was to determine whether KI reduces the discoloration caused by SDF.

## 2. Materials and Methods

### 2.1. Specimen Preparation

Healthy bovine incisors without caries, discoloration, or structural defects were used as specimens (*n* = 60). The sample size calculation was performed using G*Power 3.1 (Franz Faul, Universitat Keil, Keil, Germany). Based on a previous study [18], to estimate an effect size of 0.4 with α  =  0.05, the total sample size computed was 64 for 4 groups with 0.80 power for F-tests (ANOVA). The bovine teeth were stored in a freezer before being used. The dried specimens were then immersed in deionized water for 30 min [19]. Using a low-speed handpiece (Saeshing, Seoul, Korea) with a diamond disc (NTI-Kahla, Kahla, Germany), all specimens were sectioned (10 mm × 5 mm × 5 mm) from the labial surface of the incisors (Figure 1). Subsequently, the specimens were embedded in acrylic resin (Ortho-Jet, Lang Dental Manufacturing, Wheeling, IL, USA). The surface was polished with abrasive disc paper (SiC Sand paper, R&B Inc., Daejeon, Korea) gradually using sizes of 600, 800, 1000, and 1200 grit. The schematic of the experimental procedures used in this study is shown in Figure 1.

### 2.2. Demineralization and Remineralization Procedure

For the reference area, acid-resistant nail varnish (Mix-nails, Mix & Match, Incheon, Korea) was applied in the middle of each specimen surface before generating artificial caries lesions. Two windows (windows A and B, Figure 1) were formed with a size of 3.5 mm × 5 mm. The specimens were immersed in 40 mL of a demineralization gel for 120 h at 37 °C. The composition of the demineralization gel included 2% Carbopol (Carbopol^®^ ETD 2050 polymer, Noveon Inc., Wickliffe, OH, USA), 0.1 M lactic acid gel with pH 4.8, and hydroxyapatite (calcium phosphate tribasic, Sigma, St. Louis, MO, USA), which constituted 50% of the gel [20].

After demineralization, the specimens were randomly distributed among four experimental groups (*n* = 15) (Figure 2). Each specimen was gently dried, and the experimental materials for remineralization (SDF, SDF with KI, NaF varnish) were applied according to the manual (Table 1) [21]. After 24 h, acetone was used to wipe off the previous materials, avoiding touching the enamel surface [22,23], and the materials were reapplied. The above procedure was repeated five times for 120 h (once a day for 5 days). After the procedure, the specimens were stored in artificial saliva (gastric mucin (0.22%), NaCl (0.038%), CaCl_2_·2H_2_O (0.0213%), and KH_2_PO_4_ (0.0738%) in 1000 mL distilled water (pH = 7). Table 1 shows the materials used in this research. 

### 2.3. Assessment of Enamel Demineralization and Remineralization

#### 2.3.1. Colorimetric Analysis

Images of each specimen (*n* = 60) were captured with a general-purpose DSLR camera (model 550D, Canon, Tokyo, Japan) using the following settings: shutter speed of 1/45 s, aperture value of 13.0, and ISO speed of 1600. The distance between the specimen and the camera was kept constant. Then, RGB values were extracted from the images of the total surface area of the windows using ImageJ Image Analysis Software. Delta E (ΔE) was calculated following the formula (http://colormine.org/delta-e-calculator, accessed on 1 December 2021) with RGB values. ΔE is defined as the difference between two colors, corresponding to demineralization and remineralization, in the CIE L* a* b* color space.

#### 2.3.2. Surface Microhardness Measurement

Every sixty specimens were used for surface microhardness evaluation immediately after the removal of artificial saliva at room temperature. VHN was measured at baseline, after demineralization and after remineralization. Three Vickers microhardness (VHN) readings were averaged at the middle point of the different three areas (window A, sound, and window B) with Vickers microhardness tester (MMT-X, Matsuzawa, Akita, Japan) under a 200 g load for 15 s [18] and calculated under VHN = 1854.4 P/d^2^, where d was the length of the indentation measure in millimeters.

#### 2.3.3. Polarized Light Microscopy

Five specimens from each group were randomly selected and used for the polarized light microscopy histological analysis in each experimental group. After completion of the fluoride treatment, the specimens were cut perpendicular to the treated surfaces. A 300 μm microblock was sectioned (Tech-Cut 4, Rancho Dominguez, CA, USA) and abraded using 800, 1000, and 1200 grit disc paper (SiC Sand Paper, R&B Inc., Daejeon, Korea) until a thickness of 100 μm was attained. Glass slides were used to mount the specimens on it. The slabs were submerged in deionized water, and images were acquired using polarized light microscopy (PLM, CX31-P, Olympus, Tokyo, Japan) at magnifications of 100× and 400×. The PLM images revealed the histological characteristics and the depth of demineralization and remineralization.

#### 2.3.4. Scanning Electron Microscopy

Scanning electron microscopy (SEM) for surface analysis was performed with a field-emission scanning electron microscope operating at 15 kV using randomly selected five specimens from each group. After drying the specimens with a freeze dryer, an ion coater (E-1010, Hitachi, Tokyo, Japan) was used to coat platinum on them to a thickness of 100 nm, following which the SEM observations were carried out (S-3000N, Hitachi, Japan). The images were obtained at 8000× magnification.

### 2.4. Statistical Analysis

The mean of the measured ΔE and VHN values of each group was calculated. To assess whether the change in ΔE and VHN differed across 4 groups (i.e., control, SDF/KI, SDF, and NaF groups), one-way ANOVA tests followed by Dunnett’s test were conducted. We obtained commercially available research bovine tooth specimens and randomly assigned them into these four groups. Therefore, tooth samples were assumed to be unrelated. All statistical analyses were carried out using SAS version 9.4 (SAS Institute Inc., Cary, NC, USA), and a *p*-value < 0.05 was considered significant.

## 3. Results

### 3.1. Colorimetric Analysis

The 38% SDF group recorded the largest color change values in the remineralized enamel group compared to the 38% SDF/KI, 5% NaF, and control groups (Figure 3). The ΔE values in the SDF group were distinctly different. Compared with the control group and NaF, the SDF/KI group showed very slight color change, but there was no significant difference between the groups (Figure 3).

### 3.2. Microhardness Assessment

The mean microhardness of all baseline specimens (*n* = 60) was 283.36 HV. After 120 h of demineralization, the VHN decreased to 70.32 HV. After remineralization, there was a significant difference in the surface microhardness between the experimental and control groups. The fluoride-containing group including SDF showed a higher intensity than the control (Figure 4). The groups that were remineralized with SDF and SDF/KI exhibited improved microhardness by 10% compared to the demineralized state, whereas the NaF group exhibited improved microhardness by 5.5%.

### 3.3. Polarized Light Microscopy Assessment

Representative PLM images are shown in Figure 5. The mean lesion depth after de-mineralization after 120 h was 641.1 μm. The histological features in the PLM images of the lesions show that the experimental materials are promising for enamel remineralization recovery. Although the control group was in artificial saliva, the PLM images did not show any remineralization. The remineralization effect of the SDF/KI group (156.88 μm) was higher than that of the NaF group (145.8 μm). The color shade of the SDF group was too dark, making it impossible to measure with PLM. The lesion depths significantly reduced after fluoride treatment. Our result confirmed that remineralization commences from the outside and proceeds toward the inside.

### 3.4. Scanning Electron Microscopy Assessment

The process of demineralization followed by remineralization was qualitatively characterized by observing the specific morphological and structural features of the enamel surface in their SEM images (Figure 6). After 120 h of remineralization, different enamel surface forms were observed in the four groups. The enamel surface of the control group exhibited more porosity than those of the other groups (Figure 6a). The surface was relatively smoother and denser in the SDF/KI and SDF groups (Figure 6b,c) compared to NaF group (Figure 6d).

## 4. Discussion

The present study evaluated the potential effects of 38% SDF/KI, 38% SDF, and 5% NaF varnish to treat artificial enamel caries lesions. The application of 38% SDF/KI led to no significant difference in enamel remineralization compared to the application of 38% SDF in enamel caries and was more effective than the application of 5% NaF. In addition, there was no significant difference in color between the control and NaF group in the SDF/KI group, indicating that KI reduced the color change.

From a pharmacokinetics perspective, SDF has been shown to be a generally safe and biocompatible topical treatment [21] with high effectiveness in caries prevention and arresting [22]. According to Braga’s study, SDF was not only effective but also fast in arresting dental caries in partially erupted permanent molars pit and fissure caries [23]. It has also been found that SDF are useful to treat tooth hypersensitivity, induce dentine desensitization and as a disinfectant during root canal treatment [24]. In addition, it is easy to use, has a relatively short treatment time, is economical, and has a wide application range, from enamel-limited incipient dental caries to moderate caries progressed to dentin, unlike other fluoride agents [25].

The most significant disadvantage of SDF is the aesthetic concern caused by tooth surface discoloration [7]. In this study, the uses of KI after SDF reduced the discoloration. This is because KI reacts with the excess free ions to form silver iodide, which manifests as a white color [26]. The use of a saturated KI solution immediately after SDF application was suggested as a strategy to resolve the issue of discoloration caused by SDF [13]. Many in vitro studies have revealed the effectiveness of KI in reducing discoloration [27]. Clinical studies in primary dentition have also demonstrated aesthetic improvements [28]. However, the effect of SDF/KI on the tooth color is still disputed; a clinical trial reported that tooth discoloration still occurs in adult root cavities [16].

The spread of caries, i.e., whether the caries has affected only the enamel or has progressed to dentin, could influence discoloration. In vitro studies on a human dentin block after SDF/KI remineralization still reported color change compared to adjacent restoration [29], whereas in our study, the measured ΔE value indicates significant discoloration with the SDF group and limited color difference in the SDF/KI group. The previously reported clinical trial studies were mainly conducted on caries that had extended to dentin. More studies where the lesions are limited to the enamel are needed.

Among the many methods to evaluate the remineralization of enamel, Vickers’ method and the Knoop hardness are used to measure the microhardness of the tooth surface [30]. Vickers’ method is mainly used because the resulting pyramid-shaped indent is easy to measure, as well as detect visually and digitally [31]. After remineralization, there was a significant difference in the surface microhardness between the fluoride-containing group and control group, confirming the effect of fluoride. Calcium fluoride (CaF_2_) and silver phosphate (Ag_3_PO_4_) are representative products after applying SDF to dental minerals [32]. Fluoride is a method that promotes remineralization of early enamel lesions by forming fluorapatite [33]. Preservation of Ca^2+^ and PO_4_^3−^ results in remineralization due to increased mineral content of the lesion [34]. Remineralization was effective in the SDF/KI and SDF groups, followed by the NaF group. The important point of this result is that the effect of remineralization of enamel did not decrease even with the inclusion of KI. However, the increase of microhardness in SDF-containing groups has another factor. M. Akyildiz and I.S. Sonmez reported that silver oxide and silver iodide deposition could affect mineral density and increase microhardness of enamel and dentin [35]. Moreover, this protective layer prevents further loss of calcium and phosphate in the demineralized enamel [36].

PLM analysis was performed simultaneously to evaluate the depth of demineralization and remineralization. The depth of the remineralized enamel surface with increasing mineral content can be easily evaluated by the PLM images [37]. The SDF/KI group exhibit increased depths of remineralization compared to the control and NaF groups. Moreover, SEM analysis of the SDF-treated tooth revealed better remineralization, which has also been reported before [38]. The higher mineral density and remineralization rate in the SDF groups can be attributed to the existence of calcium fluoride and silver phosphate [34]. The hydroxyl ions of hydroxyapatite are displaced with fluoride ions to form fluorapatite. Moreover, silver phosphate acts as a phosphate ion reservoir, which aids in the formation of fluorapatite [39]. The high concentration of fluoride and silver ions from SDF increases the remineralization depth and quality.

According to ADA Evidence-based Dentistry Clinical Practice Guideline, experts have recommended 2.26% NaF varnish and 1.23% APF gel for professional fluoride application [31]. Although the remineralization ability of SDF is excellent, it has been reluctant to use it for discoloration. This study implies that SDF/KI, with its high fluoride efficacy and less coloring, could be a superior alternative for the prevention and inhibition of incipient caries, offering value as an alternative to routine fluoride application treatments.

Although there are some differences between the human and bovine teeth, it is reported that bovine teeth are suitable for replacing human teeth in comparative demineralization/remineralization experiments [40,41]. To overcome the differences in bovine enamel such as faster demineralization in enamel and chemical composition [42], the demineralization and remineralization procedure was carried out for a short time (120 h). In addition, the specimens were kept in artificial saliva solution, to reproduce the natural effect of remineralization by saliva in the oral cavity [43]. However, in the future, more detailed controlled follow-up studies through research on the human body are needed.

## 5. Conclusions

In conclusion, SDF is an effective, efficient, inexpensive, and easily applicable agent. SDF/KI was also effective for dental enamel remineralization even with the inclusion of KI, which reduces discoloration. SDF/KI showed higher remineralization (though not significant) in comparison to NaF. Long-term clinical trials are needed to confirm the superiority of these remineralized materials, but we suggest that SDF/KI is an acceptable and practical alternative agent for caries prevention and enhancement.

## Figures and Tables

**Figure 1 materials-15-04523-f001:**
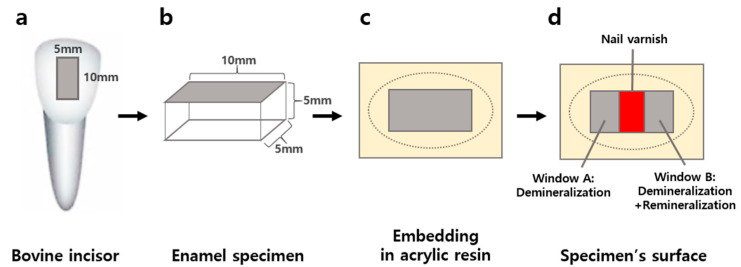
Specimen preparation. (**a**) Frontal view of bovine tooth and the area used for the specimens. (**b**) Obtained specimens (10 mm × 5 mm × 5 mm). (**c**) Specimens embedded in acrylic resin. (**d**) Expected experimental outcome using the specimens.

**Figure 2 materials-15-04523-f002:**
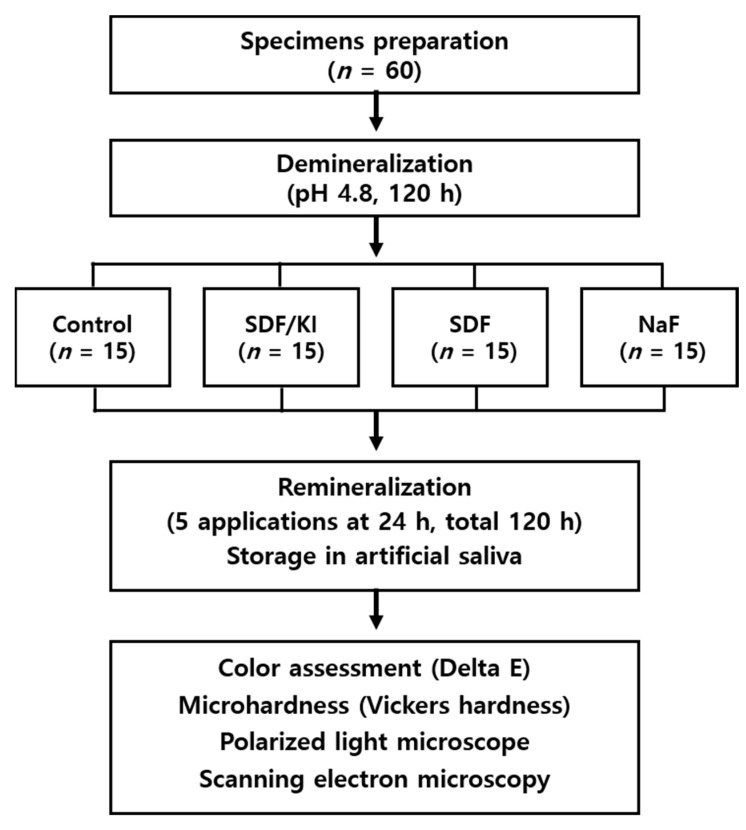
Flow chart.

**Figure 3 materials-15-04523-f003:**
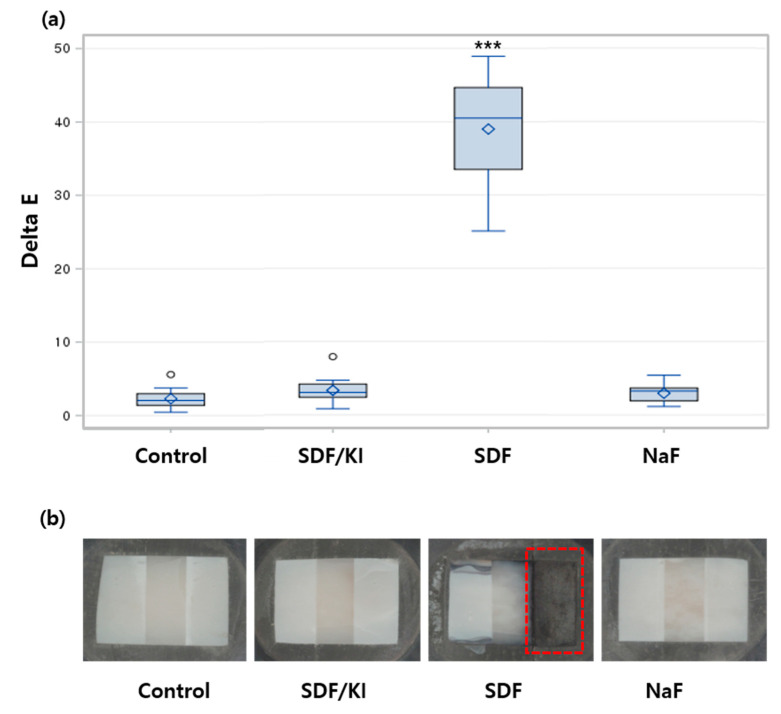
Delta E assessment of enamel surfaces after demineralization and remineralization. (**a**) The discoloration was significant in SDF (red dashed line), and there were no differences among the control, SDF/KI, and NaF groups. *** *p* < 0.001, one-way repeated measures ANOVA and Dunnett’s multiple comparisons test. (**b**) Image captured after remineralization. Critical dark discoloration was observed in the SDF group.

**Figure 4 materials-15-04523-f004:**
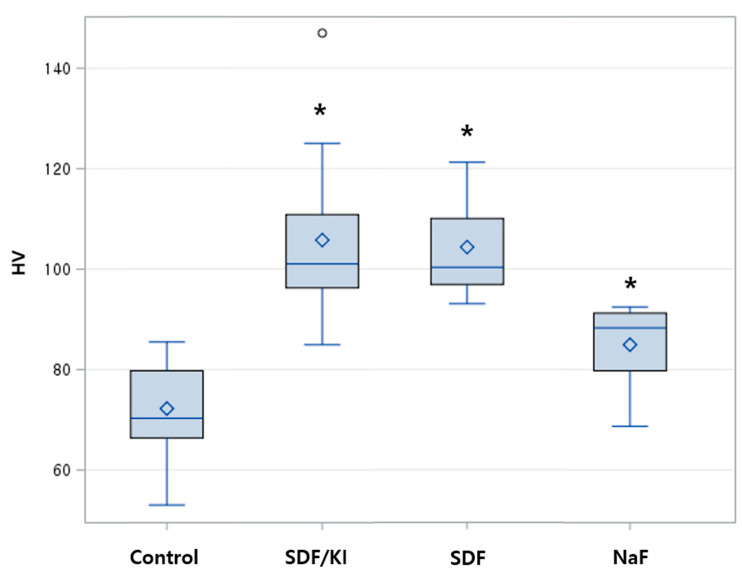
Vickers microhardness (HV) of enamel surfaces after 120 h of demineralization followed by remineralization. One-way ANOVA was conducted with * *p* < 0.05.

**Figure 5 materials-15-04523-f005:**
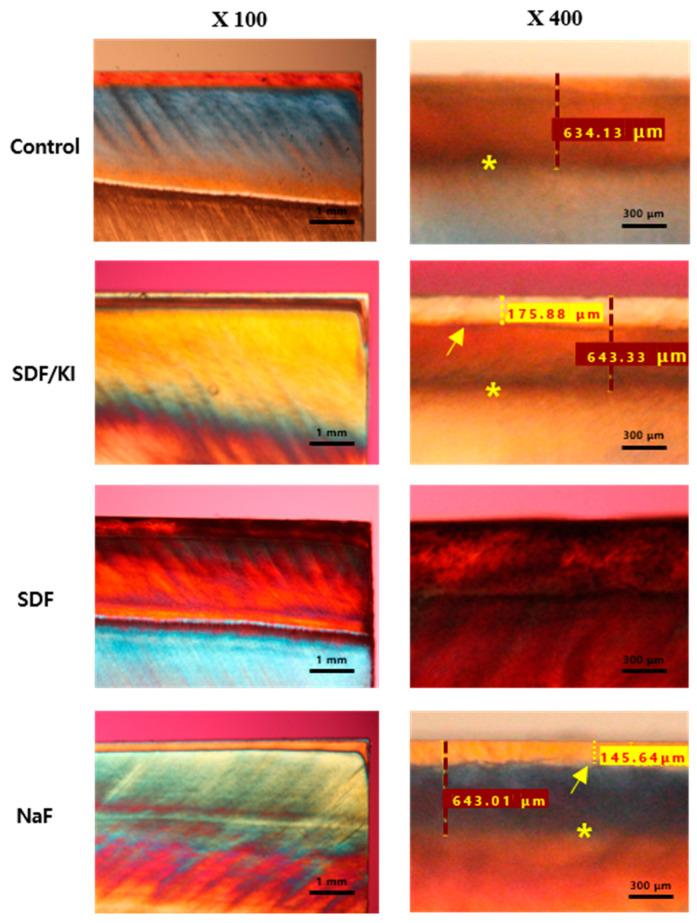
Representative polarization light micrograph images of post-demineralization and remineralization at magnifications of 100× and 400×. *: Margin of demineralization; arrow: margin of remineralization; red dashed line: the depth of demineralization; yellow dashed line: the depth of remineralization. The border of the SDF group could not be measured due to discoloration.

**Figure 6 materials-15-04523-f006:**
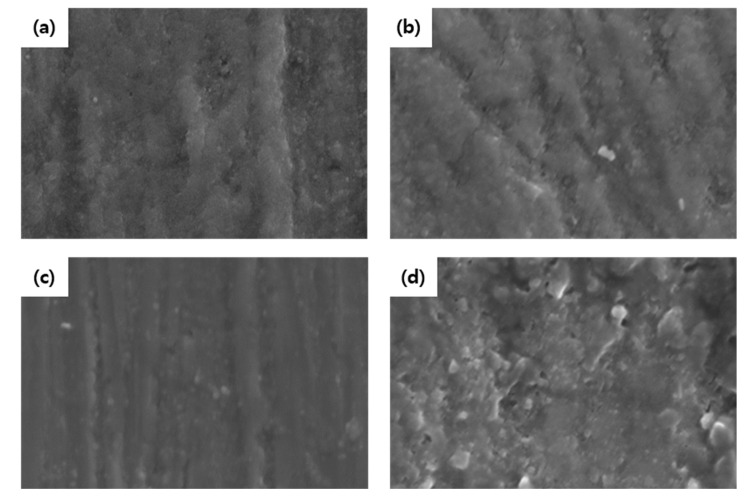
Scanning electron micrographs of enamel surfaces after remineralization. (**a**) Control group; (**b**) SDF/KI; (**c**) SDF; (**d**) NaF (magnification: 8000×).

**Table 1 materials-15-04523-t001:** Description of the materials used in this study.

Group	Manufacturer	Composition	Fluoride Content, ppm F^-^	Application
SDF/KI	Riva Star (SDI, Bayswater, Australia)	Step 1: 30–35% silver fluoride and <60% ammonia solution. Step 2: Saturated KI solution.	44,800	After air-drying the demineralized surfaces, one drop of SDF was applied with a micro-brush for 1 min. Subsequently, a saturated KI solution was applied until the creamy-white precipitates turned clear. The area was rinsed with large volumes of distilled water for 30 s.
SDF	Saforide (Toyo Seiyaku, Kasei Ltd., Osaka, Japan)	38% SDF.	44,800	After air-drying the demineralized surfaces, one drop of 38% SDF was added to a mixing well and applied with a micro-brush to the enamel surfaces for 1 min. After 2 min, the specimens were rinsed with distilled water for 30 s.
NaF	Clinpro (3M ESPE, St Paul, MN, USA)	5% NaF.	22,600	Specimens were air-dried and 5% NaF was applied on the demineralized surfaces using a micro-brush for 20 s.

Abbreviations: SDF: silver diamine fluoride; KI: potassium iodide.

## Data Availability

Not applicable.

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
