# Peer review of "Effect of Silver Diamine Fluoride and Potassium Iodide Solution on Enamel Remineralization and Discoloration in Artificial Caries"

_materials, 2022, doi:10.3390/ma15134523_

Round 1

Reviewer 1 Report

This study (Effect of Silver Diamine Fluoride and Potassium Iodide Solution on Enamel Remineralization and Discoloration in Artificial Caries) is interesting because there is no conclusive evidence on the effect of KI on SDF remineralization efficiency on enamel carious lesions. The study design, on the other hand, has significant limitations. As far as I'm concerned, they may have an impact on the overall findings and thus the reliability of the results.

  1. Please explain how the sample size was calculated.
  2. How many samples did you use in each experiment? Please be as specific as possible.
  3. Clinically, each of the three treatment groups was used once and then repeated at 3–6 months or 1-year intervals, depending on the individual’s caries risk. Why did the authors design this study to apply materials repeatedly daily for 5 days and wipe them off with acetone before each re-application, possibly affecting the mineral content of the tooth?
  4. Under normal oral conditions, remineralization and demineralization occur continuously. As a result, most in vitro studies include a pH-cycling process to simulate the real oral condition. The pH-cycling used in studies of SDF, which has antibacterial activity, was frequently bacterial pH-cycling, which is closer to the actual oral condition and allows the observation of SDF's antibacterial effect. However, your work has a different design, so discussing the advantages and disadvantages of your study design would be beneficial.
  5. Other limitations should also be explained, including why you were unable to overcome the study's limitations and how they impacted your overall findings.
  6. Please provide more detailed information about each experiment protocol so that it can be replicated by others. For example, how was a specimen or setting prepared for colorimetric analysis as well as how was the RGB value obtained?
  7. What was the purpose of using repeated one-way ANOVA to compare delta E values between groups when delta E values are color differences between before and after treatments, and the tooth specimens in each group are unrelated?

Author Response

  1. Please explain how the sample size was calculated.

Thank you for your valuable comments. We calculated the sample size with G power and added it in the manuscript.

Before

Healthy bovine incisors without caries, discoloration, or structural defects were used as specimens (n = 60).

After

Healthy bovine incisors without caries, discoloration, or structural defects were used as specimens (n = 60). The sample size calculation was performed using G*Power 3.1 (Franz Faul, Universitat Keil, Germany). Based on previous study [18] to estimate an effect size of 0.4 with α = 0.05, the total sample size computed was 64 for 4 groups with 0.80 power for F-tests (ANOVA).

Reference) QH Zhi, et al. An in vitro study of silver and fluoride ions on remineralization of demineralized enamel and dentine. 2013 (Sample size n=9)

  1. How many samples did you use in each experiment? Please be as specific as possible.
  2. Please provide more detailed information about each experiment protocol so that it can be replicated by others. For example, how was a specimen or setting prepared for colorimetric analysis as well as how was the RGB value obtained?

Thank you for providing us the opportunity to improve the quality of the study. We added the sample size for each experiment and described the experimental information and protocol in more detail.

After

2.3.1. Colorimetric Analysis

Images of each specimen (n = 60) were captured with a general-purpose DSLR camera (model 550D, Canon, Tokyo, Japan) using the following settings: shutter speed of 1/45 s, aperture value of 13.0, and ISO speed of 1600. The distance between the specimen and the camera was kept constant. Then, RGB values were extracted from the images of the total surface area of the windows using Image J Image Analysis Software. Delta E (ΔE) was calculated following the formula (http://colormine.org/delta-e-calculator) with RGB values. ΔE is defined as the difference between two colors, corresponding to demineralization and remineralization, in the L* a* b* color space.

2.3.2. Surface Microhardness Measurement

Every sixty specimens were used for surface microhardness evaluation immediately after the removal of artificial saliva at room temperature. VHN was measured at baseline, after demineralization and after remineralization. Three Vickers microhardness (VHN) readings were averaged at the middle point of the different three areas (window A, sound, and window B) with Vickers microhardness tester (MMT-X, Matsuzawa, Akita, Japan) under a 200 g load for 15 s [18] and calculated under VHN = 1854.4 P/d2, where d was the length of the indentation measure in millimeters.

2.3.3. Polarized Light Microscopy

Five specimens from each group were randomly selected and used for the Polarized Light Microscopy histological analysis in each experimental group. After completion of the fluoride treatment, the specimens were cut perpendicular to the treated surfaces. A 300 μm microblock was sectioned (Tech-Cut 4, Rancho Dominguez, CA, USA) and abraded using 800, 1000, and 1200 grit disc paper (SiC Sand Paper, R&B Inc., Daejeon, South Korea) until a thickness of 100 μm was attained. Glass slides were used to mount the specimens on it. The slabs were submerged in deionized water, and images were acquired using polarized light microscopy (PLM, CX31-P, Olympus, Tokyo, Japan) at magnifications of 100× and 400×. The PLM images revealed the histological characteristics and the depth of de-mineralization and remineralization.

  1. Clinically, each of the three treatment groups was used once and then repeated at 3–6 months or 1-year intervals, depending on the individual’s caries risk. Why did the authors design this study to apply materials repeatedly daily for 5 days and wipe them off with acetone before each re-application, possibly affecting the mineral content of the tooth?         

Thank you for your precious comments. I split your question into 3 answers.

3-1. In clinical practice, it is correct to measure according to the period, as you suggested. We are conducting a demineralization/remineralization experiment, in in vitro, which are conducted in a short period. The reason is also added in discussion

Discussion last paragraph

Although there are some differences between the human and bovine teeth, it is reported that bovine teeth were proper as a substitute for human teeth in comparative demineralization/remineralization experiments [35,36]. To overcome the differences in bovine enamel such as faster demineralization in enamel and chemical composition [37], demineralization and remineralization procedure was carried out for a short time (120 h). In addition, the specimens were kept in artificial saliva solution, to reproduce the naturally effect of remineralization by saliva in the oral cavity [38]. However, in the future, more detailed controlled follow-up studies through research on the human body are needed.

3-2. apply materials repeatedly daily for 5 days à This is to give the time for remineralization as much as the time for demineralization.

3-3. “wipe them off with acetone before each re-application, possibly affecting the mineral content of the tooth? “ à This methodology is commonly used for varnish analysis.

Reference 1) Borges, et al. Erosion protection by calcium lactate/sodium fluoride rinses under different salivary flows in vitro. Caries Research, 2014.

Reference 2) Magalhães, et al. Effect of a single application of TiF4 varnish versus daily use of a low-concentrated TiF4/NaF solution on tooth Erosion prevention in vitro. Caries Research, 2016.

Reference 3) Melo Alencar, et al. Morphological and chemical effects of in-office and at-home desensitising agents containing sodium fluoride on eroded root dentin. Archives of Oral Biology, 2020. “The varnish was applied in a thin layer using a micro brush, and the specimens were stored in artificial saliva. After 6 h, the varnishes were carefully removed with acetone solution (1:1) and cotton”

Reference 4) Comar, et al. TiF4 and NaF varnishes as anti-erosive agents on enamel and dentin erosion progression in vitro. 2015 “The varnishes were applied in a thin layer using a microbrush, and the specimens were stored in artificial saliva. After 6 h, the varnishes were carefully removed with acetone solution (1:1) and a scalpel blade,”

Before

After 24 h, acetone was used to wipe off the materials that were applied previously, and the materials were reapplied.

After

After 24 h, acetone was used to wipe off the materials, avoiding touching the enamel surface [19,20], and the materials were reapplied.

Moreover, acetone is a mainly used solvent in dental adhesives (Irmark, et al. Solvent type influences bond strength to air or blot-dried dentin. BMC Oral Health, 2016.). It is known that acetone is used for evaporation of water, but do not changing the mineral contents.

  1. Under normal oral conditions, remineralization and demineralization occur continuously. As a result, most in vitro studies include a pH-cycling process to simulate the real oral condition. The pH-cycling used in studies of SDF, which has antibacterial activity, was frequently bacterial pH-cycling, which is closer to the actual oral condition and allows the observation of SDF's antibacterial effect. However, your work has a different design, so discussing the advantages and disadvantages of your study design would be beneficial.

Thank you for your precious comments. Indeed, pH-cycling process is closer to the actual oral condition to confirm demineralization/remineralization. However, we focused more to examine the remineralization effect of SDF itself rather than the antibacterial activity function. Antibacterial activity through pH-cycling increases the remineralization effect of SDF, which can mimic the purpose to be identified through our experiments.

Reference 1) S. Savas, et al. Evaluation of the remineralization capacity of CPP-ACP containing fluoride varnish by different quantitative methods. J Appl Oral Sci. 2016

Reference 2) Kim, et al. Effect of Bioactive Glass-Containing Light-Curing Varnish on Enamel Remineralization. Materials, 2021.

Reference 3) Punyanirun et al. Silver diamine fluoride remineralized artificial incipient caries in permanent teeth after bacterial pH-cycling in-vitro. J Dent. 2018 Feb;69:55-59.

Reference 4) Sorkhdini et al. The effect of silver diamine fluoride in preventing in vitro primary coronal caries under pH-cycling conditions. Arch Oral Biol. 2021 Jan;121:104950.

  1. Other limitations should also be explained, including why you were unable to overcome the study's limitations and how they impacted your overall findings.

Thanks for the valuable comments. We revised the last paragraph as per your advice.

Before

However, this in vitro study has the following limitations: First, the specimens were obtained from bovine teeth rather than from natural teeth. Second, the lesions were not those caused by natural caries, but were created artificially. Third, it was difficult to reflect low levels of salivary proteins, fewer bacteria in artificial salivary solutions, and exposure to highly acidic environments, unlike the oral environments in in vitro studies. In the future, follow-up studies through research on the human body are needed.

After

Although there are some differences between the human and bovine teeth, it is reported that bovine teeth were proper as a substitute for human teeth in comparative demineralization/remineralization experiments. To overcome the differences in bovine enamel such as faster demineralization in enamel, demineralization and remineralization procedure was carried out for a short time (120 hours). In addition, the specimens were kept in artificial saliva solution, to reproduce the naturally effect of remineralization by saliva in the oral cavity. However, in the future, more detailed controlled follow-up studies through research on the human body are needed.

  1. What was the purpose of using repeated one-way ANOVA to compare delta E values between groups when delta E values are color differences between before and after treatments, and the tooth specimens in each group are unrelated?

Thank you for your valuable comment. We conducted one-way ANOVA to assess whether the change between color before and after treatment differed across 4 groups (i.e., control, SDF/KI, SDF, and NaF groups). And we obtained commercially available research bovine tooth specimens and randomly assigned them into four groups. So, therefore, we assumed that they were unrelated. We agree that the description was insufficient inducing difficulties in interpretation. Therefore, we have revised the manuscript as follows:

Before

The mean of the measured ΔE and VHN values of each group was calculated. To compare the remineralization effects of the treatment materials used in each group, repeated calculations through one-way ANOVA tests followed by Dunnett’s test were conducted. All statistical analyses were carried out using SAS version 9.4 (SAS Institute Inc.), and a p-value < 0.05 was considered significant.

After

The mean of the measured ΔE and VHN values of each group was calculated. To assess whether the change in ΔE and VHN differed across 4 groups (i.e., control, SDF/KI, SDF, and NaF groups), one-way ANOVA tests followed by Dunnett’s test were conducted. We obtained commercially available research bovine tooth specimens and randomly assigned them into these 4 groups. Therefore, tooth samples were assumed to be unrelated. All statistical analyses were carried out using SAS version 9.4 (SAS Institute Inc.), and a p-value < 0.05 was considered significant.

Reviewer 2 Report

Title: Effect of Silver Diamine Fluoride and Potassium Iodide Solution on Enamel Remineralization and Discoloration in Artificial Caries

This study's primary purpose was to evaluate the effect of 38% SDF and KI on enamel remineralization and compare it with that of the currently used 38% SDF and 5% sodium fluoride (NaF) varnish, to treat artificial caries. The secondary purpose of this study was to determine whether KI reduces the discoloration caused by SDF.

We thank the authors for their effort. However, there are many issues to be addressed as follows:

  • Main issue: This study does not provide new data or information. Many previous studies had been published in relation to this topic. For example:
  • Sorkhdini P, Crystal YO, Tang Q, Lippert F. The effect of silver diamine fluoride in preventing in vitro primary coronal caries under pH-cycling conditions. Archives of Oral Biology. 2021 Jan 1;121:104950.
  • Zhi QH, Lo EC, Kwok AC. An in vitro study of silver and fluoride ions on remineralization of demineralized enamel and dentine. Australian dental journal. 2013 Mar;58(1):50-6.
  • Sayed M, Matsui N, Hiraishi N, Nikaido T, Burrow MF, Tagami J. Effect of glutathione bio-molecule on tooth discoloration associated with silver diammine fluoride. International journal of molecular sciences. 2018 May;19(5):1322.

May the authors specify what is the new information they provide in this study?

Abstract: ok

Introduction: ok

Materials and Methods:

  • Page 3, section “2.2. Demineralization and remineralization”: Do you have a reference for the demineralization process? And did you measure the demineralization depth after the demineralization process?
  • Page 3, section “2.2. Demineralization and remineralization”: You mentioned demineralization and remineralization in the title of this section, however, you only talked about demineralizing solution!? Where is the remineralizing one?
  • Page 3, line 105: you used acetone to wipe off the nail varnish, does the acetone affect the enamel surface hardness?
  • Page 3, line 108: Remineralizing solution should be moved to the previous section “2.2. Demineralization and remineralization”. Also, do you have any references for the mineralization process?
  • Page 5, Section “2.4.1. colorimetric analysis”: The authors used a general-purpose DSLR camera. Is it accurate for this kind of test? It was better to use a colorimeter or spectrophotometer.
  • Page 5, Section “2.4.1. colorimetric analysis”: How many points were analyzed from the enamel surface in each specimen?
  • Page 5, line 130: add a reference for the load and time used in the microhardness test.

Results:

  • Page 5, line 153: The authors should divide the results section into sub-sections according to the test used.
  • Page 6, line 166: what is the “sound group”, Did you add all the VHN measurements of the sound part from all specimens together?

VHN of each specimen should be evaluated and compared to the tested part individually (compare the sound part to the tested part individually in each specimen)

  • Page 7, lines 176-177: Polarized light microscope may show the histological structure but cannot evaluate the enamel remineralization accurately. As you also mentioned that it was not possible to measure the SDF specimens. Therefore, PLM is not the right tool to use.
  • Page 8, lines 188-195: The authors talking about interprismatic regions (refer to these regions in the images)
  • Page 8, lines 188-195: SEM images are not clear, and “unlike” other previous studies, this study did not show any surface depositions in SDF or SDF/KI groups. It’s better to add SEM images at a lower magnification to show the general effect on enamel.

Discussion:

  • Generally, the discussion needs a more scientific explanation of the interactions of the tested materials with enamel structure.
  • Page 10, line 236: The authors mentioned “SEM showing blocked tubules”, Did the authors evaluate enamel or dentin? enamel does not have tubules.
  • Page 10, line 238: how about other main surface depositions? Such as silver oxide and silver iodide… This will definitely affect the mineral density and consequently affect the microhardness values; this may lead to a remineralization-like effect but it’s not actually remineralization.

Author Response

Response to the Editor and Reviewer for Manuscript: materials-1723751

Reviewer #2 comments:

Thank you for providing us the opportunity to improve the quality of the study. The reviewers’ comments are all helpful. We have considered each reviewer’s comments carefully and revised the manuscript according to these comments.

Thank you for attaching useful and qualified references. Comparing the objectives, we think that our experiments have a differentiated goal of our experiment. The difference between the attached thesis and our thesis.

1

Sorkhdini P, et al. The effect of silver diamine fluoride in preventing in vitro primary coronal caries under pH-cycling conditions. Archives of Oral Biology. 2021.

To investigate the ability of SDF, and its individual components, silver (Ag+) and fluoride (F−) ions.

2

Zhi QH, et al. An in vitro study of silver and fluoride ions on remineralization of demineralized enamel and dentine. Australian dental journal. 2013.

To compare the effect of silver fluoride, silver nitrate and potassium fluoride.

3

Sayed M, et al. Effect of glutathione bio-molecule on tooth discoloration associated with silver diamine fluoride. International journal of molecular sciences. 2018.

To evaluate the effect of Glutathione (GSH) bio-molecule on the reduction of enamel and dentin discoloration after application of 38% silver diamine fluoride solution (SDF).

(Our Study) Effect of Silver Diamine Fluoride and Potassium Iodide Solution on Enamel Remineralization and Discoloration in Artificial Caries

To evaluate the effect of 38% SDF and SDF/KI on enamel remineralization and compare it with 38% SDF and 5% sodium fluoride varnish.

Together with evaluating the discoloration of SDF and SDF/KI.

  1. Page 3, section “2.2. Demineralization and remineralization”: Do you have a reference for the demineralization process? And did you measure the demineralization depth after the demineralization process?
  2. Page 3, section “2.2. Demineralization and remineralization”: You mentioned demineralization and remineralization in the title of this section, however, you only talked about demineralizing solution!? Where is the remineralizing one?
  3. Page 3, line 108: Remineralizing solution should be moved to the previous section “2.2. Demineralization and remineralization”. Also, do you have any references for the mineralization process?

Answer 1-3) Thank you for your valuable comments. As you suggested, we changed the paragraph and added the references of our experiments.

Before

2.2. Demineralization and Remineralization

For the reference area, acid-resistant nail varnish (Mix-nails, Mix & Match, Incheon, South Korea) was applied in the middle of each specimen surface before generating artificial caries lesions. Two windows (windows A and B, Figure 1) were formed with a size of 3.5 mm × 5 mm. The specimens were immersed in 40 mL of a demineralization gel for 120 h at 37 °C. The composition of the demineralization gel included 2% Carbopol (Carbopol® ETD 2050 polymer, Noveon Inc., Wickliffe, OH, USA), 0.1 M lactic acid gel with pH 4.8, and hydroxyapatite (calcium phosphate tribasic, Sigma, St. Louis, MO, USA), which con-stituted 50% of the gel.

2.3. Fluoride Application for Remineralization

After demineralization, the specimens were randomly distributed among four experi-mental groups (n = 15) (Figure 2). Each specimen was gently dried, and the experimental materials for remineralization were applied according to the manual (Table 1). After 24 h, acetone was used to wipe off the materials that were applied previously, and the materials were reapplied. The above procedure was repeated five times for 120 h (once a day for 5 days). After the procedure, the specimens were stored in artificial saliva (gastric mucin (0.22%), NaCl (0.038%), CaCl2·2H2O (0.0213%), and KH2PO4 (0.0738%) in 1000 mL dis-tilled water, pH = 7). Table 1 shows the materials used in this research.

2.4. Assessment of Enamel Demineralization and Remineralization

After

2.2. Demineralization and Remineralization procedure

For the reference area, acid-resistant nail varnish (Mix-nails, Mix & Match, Incheon, South Korea) was applied in the middle of each specimen surface before generating artificial caries lesions. Two windows (windows A and B, Figure 1) were formed with a size of 3.5 mm × 5 mm. The specimens were immersed in 40 mL of a demineralization gel for 120 h at 37 °C. The composition of the demineralization gel included 2% Carbopol (Carbopol® ETD 2050 polymer, Noveon Inc., Wickliffe, OH, USA), 0.1 M lactic acid gel with pH 4.8, and hydroxyapatite (calcium phosphate tribasic, Sigma, St. Louis, MO, USA), which constituted 50% of the gel [20].

After demineralization, the specimens were randomly distributed among four experimental groups (n = 15) (Figure 2). Each specimen was gently dried, and the experimental materials for remineralization (SDF, SDF with KI, NaF varnish) were applied according to the manual (Table 1) [21]. After 24 h, acetone was used to wipe off the previous materials, avoiding touching the enamel surface [22,23], and the materials were reapplied. The above procedure was repeated five times for 120 h (once a day for 5 days). After the procedure, the specimens were stored in artificial saliva (gastric mucin (0.22%), NaCl (0.038%), CaCl2·2H2O (0.0213%), and KH2PO4 (0.0738%) in 1000 mL distilled water, (pH = 7). Table 1 shows the materials used in this research.

2.3. Demineralization and Remineralization procedure

  1. Page 3, line 105: you used acetone to wipe off the nail varnish, does the acetone affect the enamel surface hardness?

Thank you for the question. This methodology (using acetone for removing varnish materials) is commonly used for varnish analysis.

Reference 1) Borges, et al. Erosion protection by calcium lactate/sodium fluoride rinses under different salivary flows in vitro. Caries Research, 2014.

Reference 2) Magalhães, et al. Effect of a single application of TiF4 varnish versus daily use of a low-concentrated TiF4/NaF solution on tooth Erosion prevention in vitro. Caries Research, 2016.

Reference 3) Melo Alencar, et al. Morphological and chemical effects of in-office and at-home desensitising agents containing sodium fluoride on eroded root dentin. Archives of Oral Biology, 2020. “The varnish was applied in a thin layer using a micro brush, and the specimens were stored in artificial saliva. After 6 h, the varnishes were carefully removed with acetone solution (1:1) and cotton”

Reference 4) Comar, et al. TiF4 and NaF varnishes as anti-erosive agents on enamel and dentin erosion progression in vitro. 2015 “The varnishes were applied in a thin layer using a microbrush, and the specimens were stored in artificial saliva. After 6 h, the varnishes were carefully removed with acetone solution (1:1) and a scalpel blade,”

Before

After 24 h, acetone was used to wipe off the materials that were applied previously, and the materials were reapplied.

After

After 24 h, acetone was used to wipe off the materials, avoiding touching the enamel surface [19,20], and the materials were reapplied.

Moreover, acetone is a mainly used solvent in dental adhesives (Irmark, et al. Solvent type influences bond strength to air or blot-dried dentin. BMC Oral Health, 2016.). It is known that acetone is used for evaporation of water, but do not changing the mineral contents.

  1. Page 5, Section “2.4.1. colorimetric analysis”: The authors used a general-purpose DSLR camera. Is it accurate for this kind of test? It was better to use a colorimeter or spectrophotometer.

Thank you for the precious advice.

As you mentioned, we totally agree that shade determination may be precisely performed with spectrophotometers. However, according to Wee, et al., they concluded as follows; “Commercial SLR digital cameras when combined with the appropriate calibration protocols showed potential for use in the color replication process of clinical dentistry.” Development of digital camera and color analysis software like ImageJ allows easy and convenient colorimetric analysis without special devices. The device-dependent color images of input devices (i.e. digital cameras) were converted to a standard device-independent color space (i.e. CIE LAB and XYZ)

Reference 1) D. Anand, et al. Shade Selection: Spectrophotometer vs Digital camera – A comparative in-vitro study. 2016

Reference 2) AG. Wee, et al. Color accuracy of commercial digital cameras for use in dentistry. Dent Mater, 2005

Before

Images of each specimen (n = 60) were captured with a general-purpose DSLR camera (model 550D, Canon, Tokyo, Japan) using the following settings: shutter speed of 1/45 s, aperture value of 13.0, and ISO speed of 1600.

After

Images of each specimen (n = 60) were captured with a general-purpose DSLR camera (model 550D, Canon, Tokyo, Japan) using the following settings: shutter speed of 1/45 s, aperture value of 13.0, and ISO speed of 1600. The distance between the specimen and the camera was kept constant.

  1. Page 5, Section “2.4.1. colorimetric analysis”: How many points were analyzed from the enamel surface in each specimen?                                                   

Thank you for your very sharp and important opinion.

First, RGB values were extracted from the images of the total surface area of each window using Image J Image Analysis Software. Secondly, delta E (ΔE) was calculated following the formula (http://colormine.org/delta-e-calculator) with RGB values.

  1. Page 5, line 130: add a reference for the load and time used in the microhardness test.

Thank you for your precious comments. We added the reference for the load and time used in the microhardness test.           

Before

Vickers microhardness tester (MMT-X, Matsuzawa, Akita, Japan) under a 200 g load for 15 s. The VHN was calculated under VHN = 1854.4 P/d2, where P was 200 grams, and d was the length of the indentation measure in millimeters.

After

Three Vickers microhardness (VHN) readings were averaged at the middle point of the different three areas (window A, sound, and window B) with Vickers microhardness tester (MMT-X, Matsuzawa, Akita, Japan) under a 200 g load for 15 s [18] and calculated under VHN = 1854.4 P/d2, where d was the length of the indentation measure in millimeters.

Reference 1) QH. Zhi, et al., An in vitro study of silver and fluoride ions on remineralization of demineralized enamel and dentine. Aust Dent J 2013

  1. Page 5, line 153: The authors should divide the results section into sub-sections according to the test used.

Thank you for your valuable comments. We subdivided the result according to the test used.

Before

Results

The 38% SDF group recorded the largest color change values in the remineralized enamel group compared to the 38% SDF/KI, 5% NaF, and control groups (Figure 3). The ΔE values in the SDF group were distinctly different. Compared with the control group and NaF, the SDF/KI group showed very slight color change, but there was no significant difference between the groups (Figure 3).

The mean microhardness of the sound group (n = 60) was 283.36 HV. After 120 h of demineralization, the VHN decreased to 70.32 HV. After remineralization, there was a significant difference in the surface microhardness between the experimental and control groups. The fluoride-containing group including SDF showed a higher intensity than the control. (Figure 4). The groups that were remineralized with SDF and SDF/KI exhibited improved microhardness by 10% compared to the demineralized state, whereas the NaF group exhibited improved microhardness by 5.5%.

The histological features in the PLM images of the lesions show that the experimental materials are promising for enamel remineralization recovery (Figure 5). Although the control group was in artificial saliva, the PLM images did not show any remineralization. The remineralization effect of the SDF/KI group was higher than that of the NaF group. The color shade of the SDF group was too dark, making it impossible to measure with PLM. The lesion depths significantly reduced after fluoride treatment. Our result con-firmed that remineralization commences from the outside and proceeds toward the in-side.

The process of demineralization followed by remineralization was qualitatively characterized by observing the specific morphological and structural features of the enamel surface in their SEM images (Figure 6). After 120 h of remineralization, different enamel surface forms were observed in the four groups. The enamel surface of the control group exhibited more porosity than those of the other groups (Figure 6(a)). The surface was relatively smooth in the SDF/KI and SDF groups (Figure 6(b), (c)). The interprismatic areas were partially occluded, with exposed interprismatic patches visible on the surface of the NaF group (Figure 6(d)).

After

3.1. Colorimetric Analysis

The 38% SDF group recorded the largest color change values in the remineralized enamel group compared to the 38% SDF/KI, 5% NaF, and control groups (Figure 3). The ΔE values in the SDF group were distinctly different. Compared with the control group and NaF, the SDF/KI group showed very slight color change, but there was no significant difference between the groups (Figure 3).

3.2. Microhardness Assessment

The mean microhardness of all baseline specimens (n = 60) was 283.36 HV. After 120 h of demineralization, the VHN decreased to 70.32 HV. After remineralization, there was a significant difference in the surface microhardness between the experimental and control groups. The fluoride-containing group including SDF showed a higher intensity than the control. (Figure 4). The groups that were remineralized with SDF and SDF/KI exhibited improved microhardness by 10% compared to the demineralized state, whereas the NaF group exhibited improved microhardness by 5.5%.

3.3. Polarized Light Microscopy assessment

Representative PLM images are shown in Figure 5. The histological features in the PLM images of the lesions show that the experimental materials are promising for enamel remineralization recovery. Although the control group was in artificial saliva, the PLM images did not show any remineralization. The remineralization effect of the SDF/KI group was higher than that of the NaF group. The color shade of the SDF group was too dark, making it impossible to measure with PLM. The lesion depths significantly reduced after fluoride treatment. Our result confirmed that remineralization commences from the outside and proceeds toward the inside.

  1. Page 6, line 166: what is the “sound group”, Did you add all the VHN measurements of the sound part from all specimens together? VHN of each specimen should be evaluated and compared to the tested part individually (compare the sound part to the tested part individually in each specimen)

Thank you for your valuable comments. We agree that the description was insufficient inducing difficulties in interpretation. Therefore, we have revised the manuscript as follows:

Before

The mean microhardness of the sound group (n = 60) was 283.36 HV. After 120 h of demineralization, the VHN decreased to 70.32 HV.

After

The mean microhardness of all baseline specimens (n = 60) was 283.36 HV. After 120 h of demineralization, the VHN decreased to 70.32 HV. After remineralization, there was a significant difference in the surface microhardness between the experimental and control groups.

After

(2.3.2 Surface Microhardness Measurement) Every sixty specimens were used for surface microhardness evaluation immediately after the removal of artificial saliva at room tempera

ture. VHN was measured at baseline, after demineralization and after remineralization.

  1. Page 7, lines 176-177: Polarized light microscope may show the histological structure but cannot evaluate the enamel remineralization accurately. As you also mentioned that it was not possible to measure the SDF specimens. Therefore, PLM is not the right tool to use.

Thank you for your precious advice. First, we disagree that PLM is not the right tool. There are many previous studies measuring demineralization/remineralization of enamel using the PLM. We think that it is a commonly used technique for evaluating demineralization/remineralization of enamel.

Second, the purpose of our study was to evaluate the remineralization ability of “SDF/KI”, not SDF alone. Therefore, we thought it was meaningful to present the remineralization value of “SDF/KI”, and to compare it with NaF, the representative remineralization agent

Reference 1) R Rahan, et al. A Polarized Light Microscopic Study to Comparatively evaluate Four Remineralizing Agents on Enamel viz CPP-ACPF, ReminPro, SHY-NM and Colgate Strong Teeth, Int J Clin Pediatr Dent. 2015.

Reference 2) S Shah and P Birur. Polarized light microscopic evaluation of remineralization by casein phosphopeptide-amorphous calcium phosphate paste of artificial caries-like lesion: An in vitro study, 2015

Reference 3) FA de Godoi, et al. Remineralizing effect of commercial fluoride varnishes on artificial enamel lesions. Org Res Dent Mat, 2018.

Reference 4) K Bansal. In vivo remineralization of artificial enamel carious lesions using a mineral-enriched mouthrinse and a fluoride dentifrice: A polarized light microscopic comparative evaluation. 2010

Questions about SEM

  1. Page 8, lines 188-195: The authors talking about interprismatic regions (refer to these regions in the images)
  2. Page 8, lines 188-195: SEM images are not clear, and “unlike” other previous studies, this study did not show any surface depositions in SDF or SDF/KI groups. It’s better to add SEM images at a lower magnification to show the general effect on enamel.
  3. Page 10, line 236: The authors mentioned “SEM showing blocked tubules”, Did the authors evaluate enamel or dentin? enamel does not have tubules.

Thank you for your precious advice. We totally agree with your comments.

As a result of the review, there were existing problems in using the data presented by SEM, so we decided to exclude the SEM assessment part.

  1. Generally, the discussion needs a more scientific explanation of the interactions of the tested materials with enamel structure.

We revised the discussion as you suggested. Thank you.  

  1. Page 10, line 238: how about other main surface depositions? Such as silver oxide and silver iodide… This will definitely affect the mineral density and consequently affect the microhardness values; this may lead to a remineralization-like effect but it’s not actually remineralization.

Thank you for your valuable comments. We agree that the deposition of silver oxide and ions may increase the microhardness. That’s why we need histological findings (PLM) together to evaluate remineralization effects. We also revised the manuscript as follows:

After

Among the many methods to evaluate the remineralization of enamel, Vickers’ method and the Knoop hardness are used to measure the microhardness of the tooth sur-face [31]. Vickers’ method is mainly used because the resulting pyramid-shaped indent is easy to measure, as well as detect visually and digitally [32]. After remineralization, there was a significant difference in the surface microhardness between the fluoride-containing group and control group, confirming the effect of fluoride. Remineralization was effective in the SDF/KI and SDF groups, followed by the NaF group. The important point of this result is that the effect of remineralization of enamel did not decrease even with the inclusion of KI. However, M Akyildiz and IS Sonmez reported that silver oxide and silver iodide deposition could affect mineral density and increase microhardness of enamel and dentin. Therefore, PLM analysis was performed simultaneously to evaluate the depth of demineralization and remineralization.

The depth of the remineralized enamel surface with increasing mineral content can be easily evaluated by the PLM images [33]. The SDF/KI group exhibit increased depths of remineralization compared to the control and NaF groups.

Reference 1) M Akyildiz and IS Sonmez. Comparison of remineralising potential of nano silver fluoride, silver diamine fluoride and sodium fluoride varnish on artificial caries: an in vitro study.  Oral Health Prev Dent, 2019

Reviewer 3 Report

Dear Authors

Paper is well revised and i found very good evidence in the field of SDF. I would just suggest to include some information from below two references in your introduction and discussion part which will be great and stregthen your work. 

a) Haq, Jameela, et al. "SILVER DIAMINE FLUORIDE: A MAGIC BULLET FOR CARIES MANAGEMENT." Fluoride 54.3 (2021): 210-218.

b) Huang, Wei-Te, Saroash Shahid, and Paul Anderson. "Applications of silver diamine fluoride in management of dental caries." Advanced Dental Biomaterials. Woodhead Publishing, 2019. 675-699. 

Also, try to add few more lines on the conslusion in regards of your outcomes of the work.

Author Response

Response to the Reviewer 3 for Manuscript: materials-1723751

Thank you for providing us the opportunity to improve the quality of the study. The comments and references are all very helpful. We have revised the manuscript according to your comments.

Reviewer #3 comments:

  1. Paper is well revised and i found very good evidence in the field of SDF. I would just suggest to include some information from below two references in your introduction and discussion part which will be great and stregthen your work.

a) Haq, Jameela, et al. "SILVER DIAMINE FLUORIDE: A MAGIC BULLET FOR CARIES MANAGEMENT." Fluoride 54.3 (2021): 210-218.

b) Huang, Wei-Te, Saroash Shahid, and Paul Anderson. "Applications of silver diamine fluoride in management of dental caries." Advanced Dental Biomaterials. Woodhead Publishing, 2019. 675-699.

Thank you for your precious comments and helpful references. We confirmed closely at the references you provide us and revised our manuscript according to your suggestions.

Before

From a pharmacokinetics perspective, SDF has been shown to be a generally safe and biocompatible topical treatment [21] with high effectiveness in caries prevention and arresting [22]. In addition, it is easy to use, economical, and has a wide application range, from enamel limited incipient dental caries to moderate caries progressed to dentin unlike other fluoride agents [23]. The most significant disadvantage of SDF is the aesthetic concern caused by tooth surface discoloration [7]. In this study, the uses of KI after SDF reduced the discoloration. This is because KI reacts with the excess free ions to form silver iodide, which manifests as a white color [24].

After

From a pharmacokinetics perspective, SDF has been shown to be a generally safe and biocompatible topical treatment [21] with high effectiveness in caries prevention and arresting [22]. According to Braga’s study, SDF was not only effective but also fast in arresting dental caries in partially erupted permanent molars pit and fissure caries [23]. It has also been found that SDF are useful to treat tooth hypersensitivity, induce dentine desensitization and as a disinfectant during root canal treatment [24]. In addition, it is easy to use, relatively short treatment time, economical, and has a wide application range, from enamel limited incipient dental caries to moderate caries progressed to dentin unlike other fluoride agents [25].

The most significant disadvantage of SDF is the aesthetic concern caused by tooth surface discoloration [7]. In this study, the uses of KI after SDF reduced the discoloration. This is because KI reacts with the excess free ions to form silver iodide, which manifests as a white color [26].

After

Among the many methods to evaluate the remineralization of enamel, Vickers’ method and the Knoop hardness are used to measure the microhardness of the tooth sur-face [30]. Vickers’ method is mainly used because the resulting pyramid-shaped indent is easy to measure, as well as detect visually and digitally [31]. After remineralization, there was a significant difference in the surface microhardness between the fluoride-containing group and control group, confirming the effect of fluoride. Calcium fluoride (CaF2) and silver phosphate (Ag3PO4) are representative products after applying SDF to dental min-erals [32]. Fluoride is a method that pro-motes remineralization of early enamel lesions by forming fluorapatite [33]. Preservation of Ca2+ and PO43- results in remineralization due to increase mineral content of the lesion [34]. Remineralization was effective in the SDF/KI and SDF groups, followed by the NaF group. The important point of this result is that the effect of remineralization of enamel did not decrease even with the inclusion of KI.

  1. Also, try to add few more lines on the conslusion in regards of your outcomes of the work.Paper is well revised and i found very good evidence in the field of SDF.

Thank you for your valuable comments. We revised our conclusion as you suggested.

Before

We concluded that SDF/KI was effective for dental enamel remineralization even with the inclusion of KI, which reduces discoloration. SDF/KI showed higher remineralization (though not significant) in comparison to NaF. Long-term clinical trial studies are re-quired to confirm the superiority of these remineralizing materials.

After

 In conclusion, SDF is an effective, efficient, inexpensive and easily applicating agent. SDF/KI was also effective for dental enamel remineralization even with the inclusion of KI, which reduces discoloration. SDF/KI showed higher remineralization (though not significant) in comparison to NaF. Long-term clinical trials are needed to confirm the superiority of these remineralized materials, but we suggest that SDF/KI is an acceptable and practical alternative agent for caries prevention and enhancement.

Round 2

Reviewer 1 Report

Thank you for providing me with the revised manuscript. However, the manuscript has not been improved enough to be published in Materials.

Some suggestions are provided below:

1. The number of samples was calculated; however, the authors used a sample size that was less than what was estimated. Is this study's sample size large enough to detect a difference?

2. SDF is usually administered once and then repeated at 3–6 months or 1-year intervals in a clinical setting. Several in vitro investigations were also designed to use SDF only once before undergoing pH-cycling for 5-7 days. Why was this study required to apply materials repeatedly every day for 5 days, and did such a large amount of application make the results practically impractical?

Author Response

  1. The number of samples was calculated; however, the authors used a sample size that was less than what was estimated. Is this study's sample size large enough to detect a difference?

Thank you for your valuable comments. Yes, we used a smaller sample size than what was estimated. However, we conducted post hoc power analysis using G*Power 3 (Erdfelder, Faul, & Buchner) and determined that our sample size (n=60) is large enough to detect a difference in both of delta E (power of 99.9%; alpha = 0.05) and in microhardness (power of 99.9%; alpha = 0.05) for four group.

Healthy bovine incisors without caries, discoloration, or structural defects were used as specimens (n = 60). The sample size calculation was performed using G*Power 3.1 (Franz Faul, Universitat Keil, Germany). Based on previous study [18] to estimate an effect size of 0.4 with α = 0.05, the total sample size computed was 64 for 4 groups with 0.80 power for F-tests (ANOVA).

Delta E /Microhardness

Figure is attached in the word file.

  1. SDF is usually administered once and then repeated at 3–6 months or 1-year intervals in a clinical setting. Several in vitro investigations were also designed to use SDF only once before undergoing pH-cycling for 5-7 days. Why was this study required to apply materials repeatedly every day for 5 days, and did such a large amount of application make the results practically impractical?           

Thank you for your valuable comments. We attach reference with performing multiple remineralization.

First, as you mentioned in clinical the SDF administered once and repeated at 3–6 months or 1-year intervals. Applying at multiple intervals in clinical practice, it was implemented compressively in vitro to evaluate the maximum remineralization potential.

Second, in order to compare with other varnish, SDF was also applied several times

Reference) Venu Varma, et al. Comparative Evaluation of Remineralization Potential of Two Varnishes Containing CPP–ACP and Tricalcium Phosphate: An In Vitro Study.2019

Reviewer 2 Report

We thank the authors for resubmitting the manuscript and for their hard work. However, still, some critical issues to address.

Materials and Method:

  • Mention the demineralization depth resulted from the demineralizing gel after 120 hours
  • You need to add SEM images as these images are the confirmatory test for your results. Also, it would be better if you can support the SEM with Energy Dispersive Spectroscopy (EDS) analysis for the surface
  • Regarding color observation (colorimetric analysis), It would be better if you add a colored images for different groups to show the color difference.

Discussion:

In order to explain your results, You need to add what kind of chemical reactions occur and the products of these reactions between SDF, SDF/KI, and NaF with tooth structure.

Author Response

Thank you for providing us the opportunity to improve the quality of the study. We revised the manuscript according to these comments.

Reviewer #2 comments:

  1. Mention the demineralization depth resulted from the demineralizing gel after 120 hours.

Thank you for your valuable comments. We added the depth of after 120 h demineralization and remineralization.

After

Representative PLM images are shown in Figure 5. The mean lesion depth after de-mineralization after 120 h was 641.1μm. The histological features in the PLM images of the lesions show that the experimental materials are promising for enamel remineralization recovery. Although the control group was in artificial saliva, the PLM images did not show any remineralization. The remineralization effect of the SDF/KI group (156.88μm) was higher than that of the NaF group (145.8μm).

  1. You need to add SEM images as these images are the confirmatory test for your results. Also, it would be better if you can support the SEM with Energy Dispersive Spectroscopy (EDS) analysis for the surface

2-1. 1st Revision: The authors talking about interprismatic regions (refer to these regions in the images) / SEM images are not clear, and “unlike” other previous studies, this study did not show any surface depositions in SDF or SDF/KI groups. It’s better to add SEM images at a lower magnification to show the general effect on enamel/ The authors mentioned “SEM showing blocked tubules”, Did the authors evaluate enamel or dentin? enamel does not have tubules.

Thank you for your valuable comments and we totally agree that SEM with Energy Dispersive Spectroscopy (EDS) analysis and lower magnification images for the surface could improve our results. Unfortunately, this time it seems to be difficult to support a better image. Form the next study we will improve it in the way you suggested. Through this figure we could confirm that SDF and SDF/KI has smoother and denser surface.

Before

Figure 6. Scanning electron micrographs of enamel surfaces after remineralization. (a) control group (b) SDF/KI (c) SDF (d) NaF (Magnification: 8000×.)

3. Results

The process of demineralization followed by remineralization was qualitatively characterized by observing the specific morphological and structural features of the enamel surface in their SEM images (Figure 6). After 120 h of remineralization, different enamel surface forms were observed in the four groups. The enamel surface of the control group exhibited more porosity than those of the other groups (Figure 6(a)). The surface was relatively smooth in the SDF/KI and SDF groups (Figure 6(b), (c)). The interprismatic areas were partially occluded, with exposed interprismatic patches visible on the surface of the NaF group (Figure 6(d)).

4. Discussion

Moreover, SEM analysis of the SDF-treated tooth revealed partially blocked tubules with more hypermineralization, which has also been reported before [28].

Revision1

Removed

Revision2

Abstract

Polarized light microscopy (PLM) and scanning electron microscopy (SEM) were performed to confirm the surfaces precisely and histologically. ...... which was also confirmed by SEM.

2. Materials and Methods

2.4.4. Scanning Electron Microscopy

Scanning electron microscopy (SEM) for surface analysis was performed with a field-emission scanning electron microscope operating at 15 kV using randomly selected five specimens from each group. After drying the specimens with a freeze dryer, an ion coater (E-1010, Hitachi, Japan) was used to coat platinum on them to a thickness of 100 nm, following which the SEM observations were carried out (S-3000N, Hitachi, Japan). The images were obtained at ×8000 magnification.

Figure 6. Scanning electron micrographs of enamel surfaces after remineralization. (a) control group (b) SDF/KI (c) SDF (d) NaF (Magnification: 8000×.)

3. Results

The process of demineralization followed by remineralization was qualitatively characterized by observing the specific morphological and structural features of the enamel surface in their SEM images (Figure 6). After 120 h of remineralization, different enamel surface forms were observed in the four groups. The enamel surface of the control group exhibited more porosity than those of the other groups (Figure 6(a)). The surface was relatively smoother and denser in the SDF/KI and SDF groups (Figure 6(b), (c)) compared to NaF group (Figure 6(d)).

4. Discussion

Moreover, SEM analysis of the SDF-treated tooth revealed better remineralization, which has also been reported before [32,33]. 

  1. Regarding color observation (colorimetric analysis), It would be better if you add a colored images for different groups to show the color difference.

Thank you for your precious comments. A colored image is attached at the bottom along with Delta E's analysis in Figure 3 in the current manuscript.

  1. In order to explain your results, You need to add what kind of chemical reactions occur and the products of these reactions between SDF, SDF/KI, and NaF with tooth structure.

We added the chemical reactions of the materials and revised the discussion.

Before

Among the many methods to evaluate the remineralization of enamel, Vickers’ method and the Knoop hardness are used to measure the microhardness of the tooth sur-face [31]. Vickers’ method is mainly used because the resulting pyramid-shaped indent is easy to measure, as well as detect visually and digitally [32]. After remineralization, there was a significant difference in the surface microhardness between the fluoride-containing group and control group, confirming the effect of fluoride. Remineralization was effective in the SDF/KI and SDF groups, followed by the NaF group. The important point of this result is that the effect of remineralization of enamel did not decrease even with the inclusion of KI.

After

Among the many methods to evaluate the remineralization of enamel, Vickers’ method and the Knoop hardness are used to measure the microhardness of the tooth sur-face [28]. Vickers’ method is mainly used because the resulting pyramid-shaped indent is easy to measure, as well as detect visually and digitally [29]. After remineralization, there was a significant difference in the surface microhardness between the fluoride-containing group and control group, confirming the effect of fluoride. Fluoride is a method that pro-motes remineralization of early enamel lesions by forming fluorapatite [30]. Remineralization was effective in the SDF/KI and SDF groups, followed by the NaF group. The important point of this result is that the effect of remineralization of enamel did not decrease even with the inclusion of KI. However, the increase of microhardness in SDF containing groups has another factor. M Akyildiz and IS Sonmez reported that silver oxide and silver iodide deposition could affect mineral density and increase microhardness of enamel and dentin [31]. Moreover, this protective layer prevents further loss of calcium and phosphate in the demineralized enamel [32].

Round 3

Reviewer 2 Report

Thanks so much for the author's efforts. This manuscript has been improved and is ready to be published. 

Author Response

Reviewer #2

We all appreciate your comments and suggestions. It was very helpful.

Thank you again for giving the opportunity to improve our study.